# The integrated genomic and epigenomic landscape of brainstem glioma

Lee H. Chen [1,2,8], Changcun Pan [3,4,5,8], Bill H. Diplas [1,2], Cheng Xu[1,2,3], Landon J. Hansen[1,2], Yuliang Wu[3], Xin Chen[3], Yibo Geng[3], Tao Sun[3], Yu Sun[3], Peng Zhang[3], Zhen Wu[3], Junting Zhang[3], Deling Li[3], Yang Zhang[3], Wenhao Wu[3], Yu Wang [3], Guangyu Li[6], Jie Yang[6], Xiaoyue Wang[7], Ce Xu[6], Sizhen Wang[6], Matthew S. Waitkus [1,2], Yiping He[1,2], Roger E. McLendon[1], David M. Ashley[2], Hai Yan[1,2✉] & Liwei Zhang[3,4,5✉]

Brainstem gliomas are a heterogeneous group of tumors that encompass both benign tumors cured with surgical resection and highly lethal cancers with no efficacious therapies. We perform a comprehensive study incorporating epigenetic and genomic analyses on a large cohort of brainstem gliomas, including Diffuse Intrinsic Pontine Gliomas. Here we report, from DNA methylation data, distinct clusters termed H3-Pons, H3-Medulla, IDH, and PA-like, each associated with unique genomic and clinical profiles. The majority of tumors within H3-Pons and H3-Medulla harbors *H3F3A* mutations but shows distinct methylation patterns that correlate with anatomical localization within the pons or medulla, respectively. Clinical data show significantly different overall survival between these clusters, and pathway analysis demonstrates different oncogenic mechanisms in these samples. Our findings indicate that the integration of genetic and epigenetic data can facilitate better understanding of brainstem gliomagenesis and classification, and guide future studies for the development of novel treatments for this disease.

[1] Department of Pathology, Duke University Medical Center, Durham 27710 NC, USA. [2] Preston Robert Tisch Brain Tumor Center, Duke University Medical Center, Durham 27710 NC, USA. [3] Department of Neurosurgery, Beijing Tiantan Hospital, Capital Medical University, Nan Si Huan Xi Lu 119, Fengtai District, 100070 Beijing, China. [4] China National Clinical Research Center for Neurological Diseases, Nan Si Huan Xi Lu 119, Fengtai District, 100070 Beijing, China. [5] Beijing Key Laboratory of Brain Tumor, Nan Si Huan Xi Lu 119, Fengtai District, 100070 Beijing, China. [6] Genetron Health (Beijing) Co. Ltd, 102208 Beijing, China. [7] State Key Laboratory of Medical Molecular Biology, Center for Bioinformatics, Institute of Basic Medical Sciences, Chinese Academy of Medical Sciences, Peking Union Medical College, 100005 Beijing, China. [8] These authors contributed equally: Lee H. Chen, Changcun Pan. ✉email: hai.yan@duke.edu; zlwtt@aliyun.com

Brainstem gliomas represent a heterogeneous group of tumors that arise from the midbrain, pons, or medulla. Among these tumors, pediatric diffuse intrinsic pontine glioma (DIPG), with a median overall survival of 9–12 months[1], has been the main research focus for the past five decades due to the inoperability and resistance to chemotherapy and radiotherapy[2–6]. Approximately 80% of pediatric DIPGs harbor K27M mutations affecting *H3F3A* or *HIST1H3B/C*[1,7–12]. These K27M-mutant tumors are associated with a particularly poor prognosis[13]. In the 2016 World Health Organization (WHO) classification of CNS tumors, the term "diffuse midline gliomas, H3K27M-mutant" was introduced to represent this DIPG tumor subset[14]. However, the molecular characteristics of the non-pediatric brainstem gliomas, such as midbrain and medulla oblongata gliomas, remain poorly characterized. Research on these tumors has been challenging due to the relatively low incidence rate and the high risks related to surgical resection resulting in low tissue availability. We collected, to our knowledge, the largest and most comprehensive cohort of brainstem gliomas, encompassing all age groups and anatomic locations, including medulla, pons, and midbrain. We performed integrated whole genome sequencing, RNA sequencing, and array-based genome-wide methylation analysis to acquire a more comprehensive picture of the molecular composition of these brain tumors. Here, we report methylation-based clusters that identify brainstem glioma subsets associated with tumor location and mutation landscape. We present two distinct clusters of H3-mutant brainstem gliomas, H3-Pons and H3-Medulla, which despite their similar genetic mutations, differ not only in location, but in methylation pattern, gene expression, and prognosis.

## Results

**Patient cohort characteristics**. We collected tumor samples and matched blood from 126 patients (detailed clinical information listed in Supplementary Table 1). Tumor locations included midbrain tegmentum (11/126, 8.7%), tectum (5/126, 4.0%), pontomesencephalic junction (2/126, 1.6%), pons (38/126, 30.2%), middle cerebellar peduncle (7/126, 5.6%), pontomedullary (16/126, 12.6%), medulla (42/126, 33.3%), and midbrain-thalamus (5/126, 4.0%) (Supplementary Fig. 1; Supplementary Table 2). Patients were aged from 2 to 62 years, with a median age of 23 years. Tumors were graded based on the WHO classification and included 8.7% (11/126) WHO grade I, 41.3% (52/126) WHO grade II, 31.0% (39/126) WHO grade III, and 19.0% (24/126) WHO grade IV tumors. The majority of original histopathological diagnoses were astrocytoma (59, 46.8%), along with 27 oligoastrocytomas (21.4%), 24 glioblastomas (19.0%), 8 pilocytic astrocytomas (PA) (6.3%), 3 gangliogliomas (2.4%), 2 pilomyxoid astrocytomas (PMA) (1.6%), 1 pleomorphic xanthoastrocytoma (PXA) (0.8%), and 1 oligodendroglioma (0.8%). Among the 38 tumors located in the pons, 33 (86.8%) were diagnosed as diffuse high-grade midline gliomas, NOS (historically known as DIPG), and 5 were focal pontine tumors, most likely pilocytic astrocytomas (13.2%). To analyze the tumor genomic and epigenomic characteristics of this tumor cohort, we performed methylation microarrays ($n = 123$) and RNA sequencing (RNAseq) ($n = 75$) on tumors included in this study, and whole genome ($n = 97$) and panel sequencing ($n = 21$) on paired tumors and normal (germline) controls.

**Methylation classification reveals distinct H3 clusters correlated with tumor locations in brainstem gliomas**. DNA methylation status has been utilized for classification of brain tumors, and could assist diagnosis and prognostication[2,13,15–17]. We performed unsupervised hierarchical clustering (linkage method: WPGMA; distance: Euclidean) using the top 20,000 most variable probes (Supplementary Table 3), excluding methylation probes on sex chromosomes and common single nucleotide polymorphisms sites. This approach revealed four distinct methylation clusters (Fig. 1). The hypermethylated cluster (methylation cluster IDH) consisted primarily of tumors bearing *isocitrate dehydrogenase 1* (*IDH1*) mutations. Histone H3 mutant samples formed two different clusters associated with tumor location (methylation clusters H3-Pons and H3-Medulla) (Fig. 1). The remaining cluster (methylation cluster PA-like) consisted largely of lower grade gliomas without detectable *IDH1* or *H3* mutations. These clusters matched with the DKFZ methylation classifier by three main classes[15], "diffuse midline gliomas H3 K27M mutant", "pilocytic astrocytoma", and "IDH mutant". However, our methylation-based clustering analysis revealed that H3-mutant samples were made up of two distinct sub-clusters, which correlated with their anatomic localization in the brainstem of either the pons (methylation cluster H3-Pons) or medulla (methylation cluster H3-Medulla). Principal component analysis of the methylation array probe data using R packages and functions (ggbiplot and prcomp)[18,19] also revealed distinct groups based on the DKFZ methylation classifier (Fig. 2a; Supplementary Fig. 2a). When performing PCA specifically for whole probes from tumors within methylation cluster H3-Pons and H3-Medulla (Fig. 2b), as well as locations in methylation cluster H3-Pons and H3-Medulla (Supplementary Fig. 2b), we observed similar trends in these clusters and locations. The first principal component could explain most of the variance in the various principal component analyses performed (96.4% in Fig. 2a and 96.7% in Fig. 2b, Supplementary Fig. 2b), indicating methylation profiling could provide significant utility as a classifier of these samples. The four distinct methylation clusters, corresponding to the H3-Pons, H3-Medulla, IDH, and PA-like subtypes, could be readily identified. Analysis using Tumor Map supported these findings by demonstrating a similar distinction of clusters correlated to tumor location[20] (Fig. 2c; Supplementary Fig. 3).

The top 20,000 variable probes used above were based on methylation data from all samples, including samples from methylation clusters IDH and PA-like. To prevent potential confounding factors for those differential probes mainly for methylation clusters IDH and PA-like, we selected the top 20,000 variable probes from only the methylation clusters H3-Pons and H3-Medulla, and we performed hierarchical clustering on these cases (Supplementary Fig. 4). Heatmap analysis showed this hierarchical clustering of subclusters indeed maintained distinct clusters. Most samples from methylation cluster H3-Medulla are tumors from the medulla, or the nearby dorsal pontomedullary junction, (23/27, 85.2%) (Supplementary Table 1, Supplementary Table 2; Supplementary Fig. 1). Tumors in the methylation cluster H3-Pons are largely pontine tumors (29 out of 47, 61.7%) and 4 from the nearby middle cerebellar peduncle (8.5%). Methylation cluster PA-like primarily consisted of medullary tumors (18 out of 34, 52.9%), along with 12 tumors from midbrain regions (8 from midbrain tegmentum and 4 from tectum, 35.3%). Unlike other clusters having focused locations, tumors from cluster *IDH* were distributed across brainstem regions, 3 from the pons (20%), 3 from the middle cerebellar peduncle (20%), and 4 from the medulla (26.7%). Among all 31 DIPG tumors with methylation data, 27 of them were included in the H3-Pons cluster (87.1%), with the remaining 4 DIPG cases clustering in the IDH cluster (12.9%).

All patients in this study were of Asian ethnicity. To evaluate if these distinct H3 clusters can be found in a predominantly non-Asian population, we combined our dataset with published

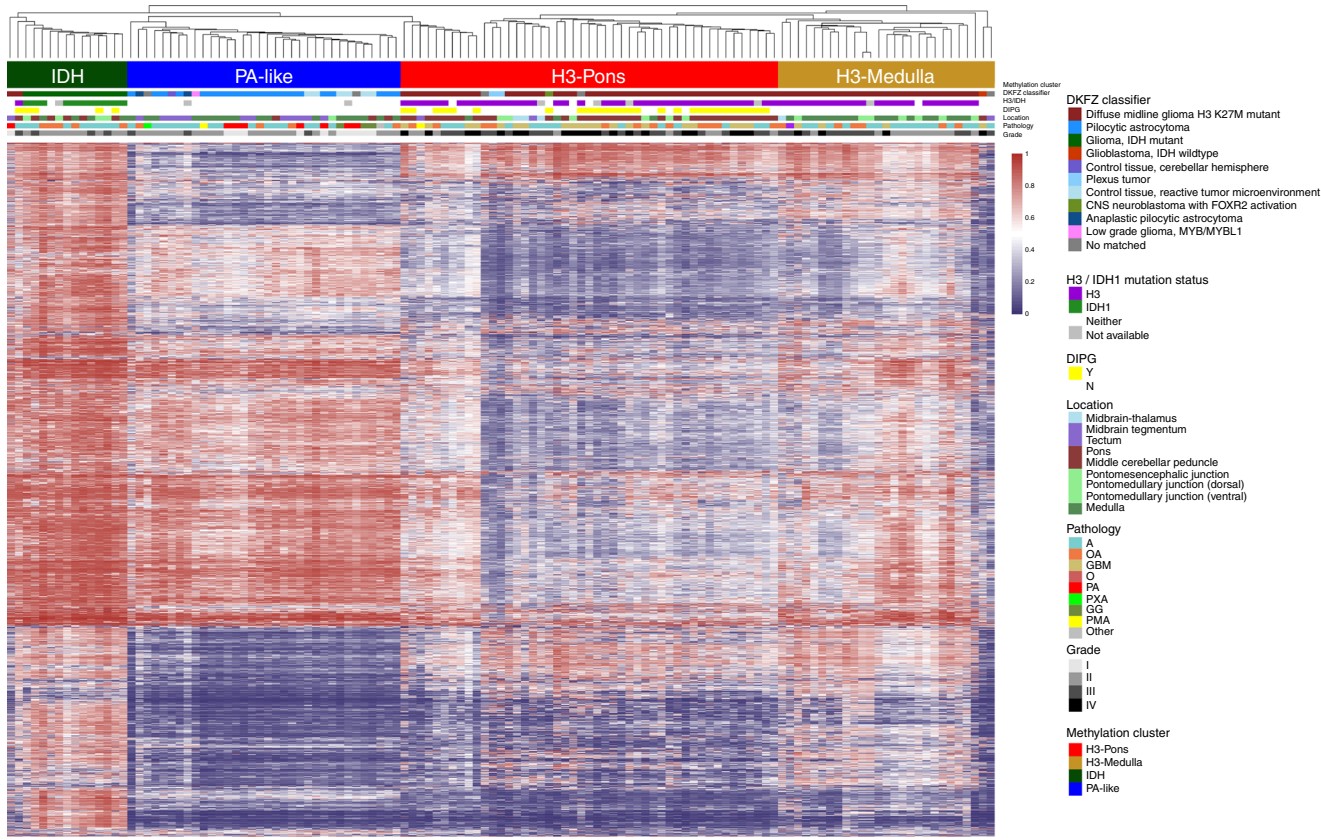

**Fig. 1 Hierarchical clustering of methylation status from top 20,000 variable probes.** Four distinct clusters could be differentiated: H3-Pons, H3-Medulla, IDH, and PA-like. Color scales indicate average beta values from methylation microarray (ranges from 0 to 1).

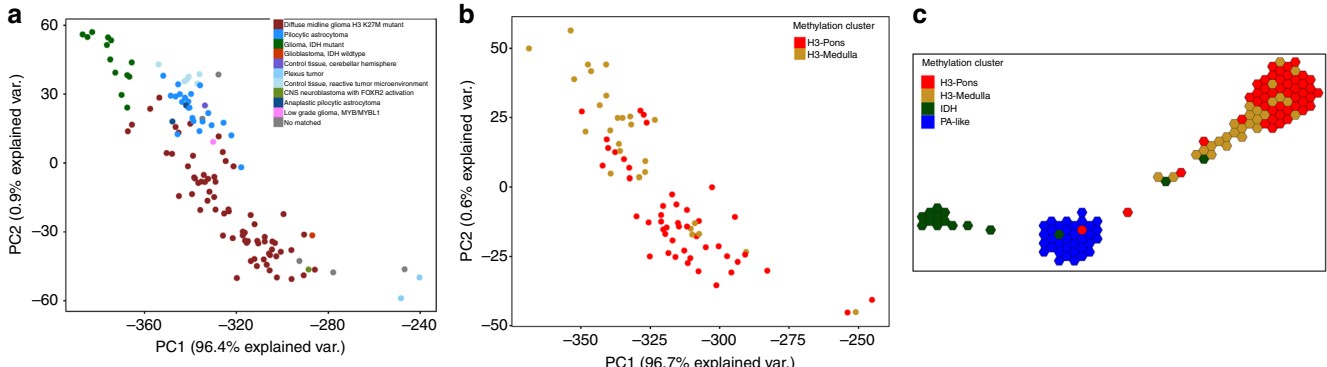

**Fig. 2 Visualization of methylation status in brainstem gliomas. a** Principal component analysis of whole probes of methylation status, colored by DKFZ classifier. **b** PCA of whole probes for tumors from H3-Medulla and H3-Pons only, colored by clusters defined in Fig. 1. **c** Tumor map from whole probes, colored by methylation clusters.

studies of 28 DIPG samples (Buczkowicz et al.[9]). From tSNE results of those selected top 20,000 variable probes, we found that those DIPG samples grouped closely with our H3-Pons samples as expected (Supplementary Fig. 5), indicating that classification according to the methylation cluster H3-Pons may be robust across ethnicities. Notably, of the 6 DIPG cases from Buczkowicz et al. that clustered toward the PA-like group, 4 were H3[WT] and all were previously classified in either the "silent" or "MYCN" methylation clusters of that study.

**Tumors of distinct methylation clusters display different genomic landscapes.** To identify somatic genetic alterations in

this brainstem glioma cohort, whole genome sequencing and panel targeted sequencing for 68 common mutated brain tumor genes were used on both the tumor samples and matched blood (Fig. 3). BWA and GATK MuTect2[21,22] were used for variant calling, and IntOgen[23,24] and FML[25] were used to predict potential driver mutations and significant noncoding region mutations. The mutation landscape of these tumors is grouped by the four distinct DNA methylation clusters defined above (Fig. 3; Supplementary Table 1). No significantly mutated noncoding regions were identified.

Methylation cluster H3-Pons and methylation cluster H3-Medulla are both enriched for H3 mutations, (H3-Pons: 40/43, 93.0%; H3-Medulla: 23/26, 88.5%), while the mutation

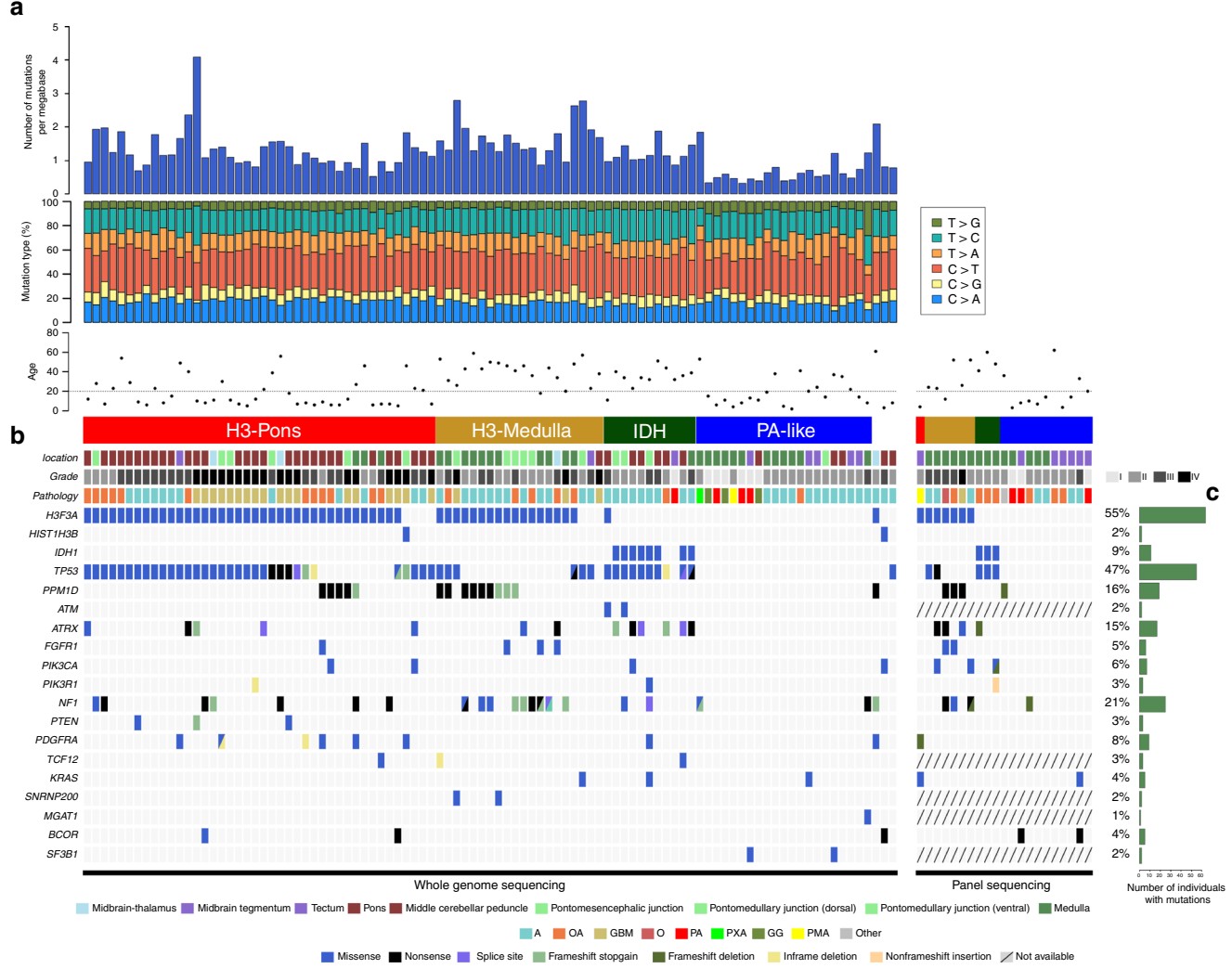

**Fig. 3 Mutation landscape of brainstem glioma samples from whole genome sequencing data. a** Numbers of mutations per megabase for each sample. **b** Clinical information and genetic alterations in brainstem glioma samples. **c** Frequency of mutations in each gene.

frequencies of *PPM1D*, *FGFR1*, and *NF1* are higher in methylation cluster H3-Medulla (*PPM1D* 46.15%, *FGFR1* 19.23%, *NF1* 46.15%) than in methylation cluster H3-Pons (*PPM1D* 11.63%, *FGFR1* 2.33%, *NF1* 16.28%) (Fig. 4a). *PPM1D* truncating mutations have been reported in brainstem gliomas, but the frequency rate varies between different studies[1,9–11,26]. Three of the 30 DIPGs harbored *PPM1D* mutations, as compared with 14 of 41 non-DIPG in H3 mutant brainstem gliomas. Methylation clusters H3-Pons and H3-Medulla shared several driver mutations, as predicated by FML/Intogen: *PPM1D, NF1, ATRX, FGFR1, H3F3A, TP53,* and *SNRNP200*. However, methylation cluster H3-Pons specifically included other distinct driver alterations, such as *BCOR, TCF12, KRAS, PTEN, MGAT1, and PIK3CA* (Fig. 4b).

As expected, methylation cluster IDH is enriched for *IDH1* mutations (78.57%) and most of these cases harbored co-occurring *TP53* (92.86%) and *ATRX* (50%) mutations. FML/Intogen analysis showed that *TP53, ATRX,* and *IDH1* were significantly mutated and potential driver mutations in this methylation cluster. Of note, the patients in this cluster are adults (age range 23–60 years).

Methylation cluster PA-like, consisting primarily of grade I or II brainstem gliomas, showed distinct patterns in DNA

methylation and genetic mutations (Figs. 1 and 3). The number of mutations for each sample was lower in samples of cluster PA-like compared with samples in other clusters (mean mutation count: 6.9; methylation cluster IDH, H3-Medulla, H3-Pons mean mutation count: 24.1). Interestingly, FML/Intogen driver analysis revealed potential driver mutations in *NF1* and *SF3B1* coding regions, and in the noncoding 3′ UTR of *EXD3*, despite their low frequency of mutations. Overall, few of the commonly associated glioma driver genes were mutated in the methylation cluster PA-like, and *NF1* and *SF3B1* were the only recurrently mutated genes in this cluster.

**Gene expression profiling reveals distinct enriched gene sets in methylation clusters H3-Pons and H3-Medulla.** We performed RNAseq on samples from the brainstem glioma cohort and evaluate patterns in gene expression (Supplementary Fig. 6). When selecting genes encoding transcription factors, similar to DNA methylation profiling, methylation clusters H3-Pons and H3-Medulla could be differentiated by gene expression profiles[27] (Supplementary Fig. 6). Next, we used HTseq[28] and edgeR[29,30] to identify differentially expressed genes between methylation clusters H3-Pons and H3-Medulla, followed by enrichment analysis

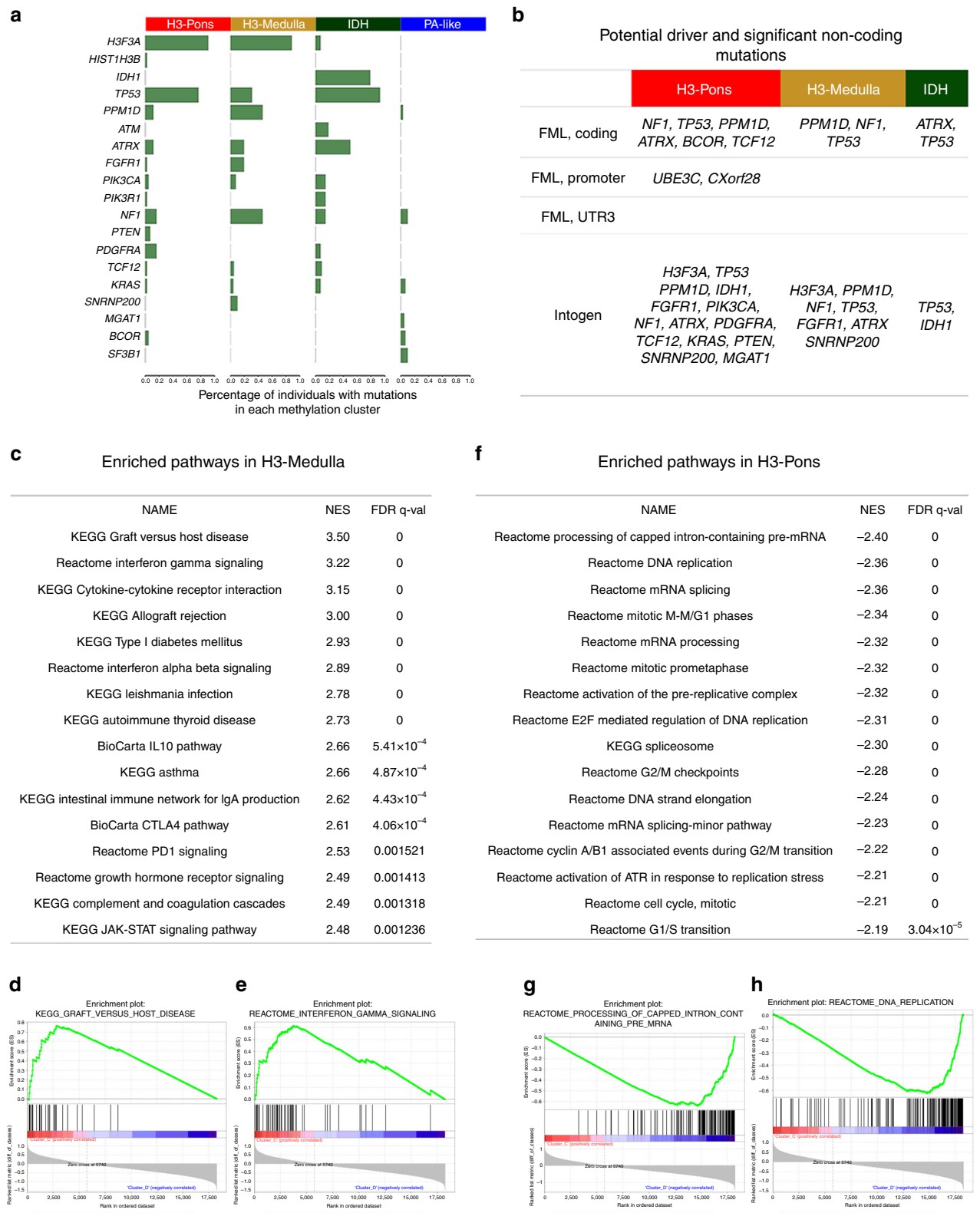

**Fig. 4 Methylation clusters differ in mutation landscapes and potential driver genes, and gene set enrichment analysis for H3-Medulla and H3-Pons. a** Frequency of mutations in each gene, grouped by clusters. **b** Potential driver and significant noncoding mutations for each cluster. **c** Enriched pathways in H3-Medulla (**d**) Enrichment Plot from H3-Medulla: KEGG Graft versus host disease (**e**) Enrichment plot from H3-Medulla Reactome Interferon gamma signaling (**f**) Enriched pathways in H3-Pons (**g**) Enrichment plot from H3-Pons: Reactome Processing of capped intron-containing pre-mRNA (**h**) Enrichment plot from H3-Pons: Reactome DNA replication.

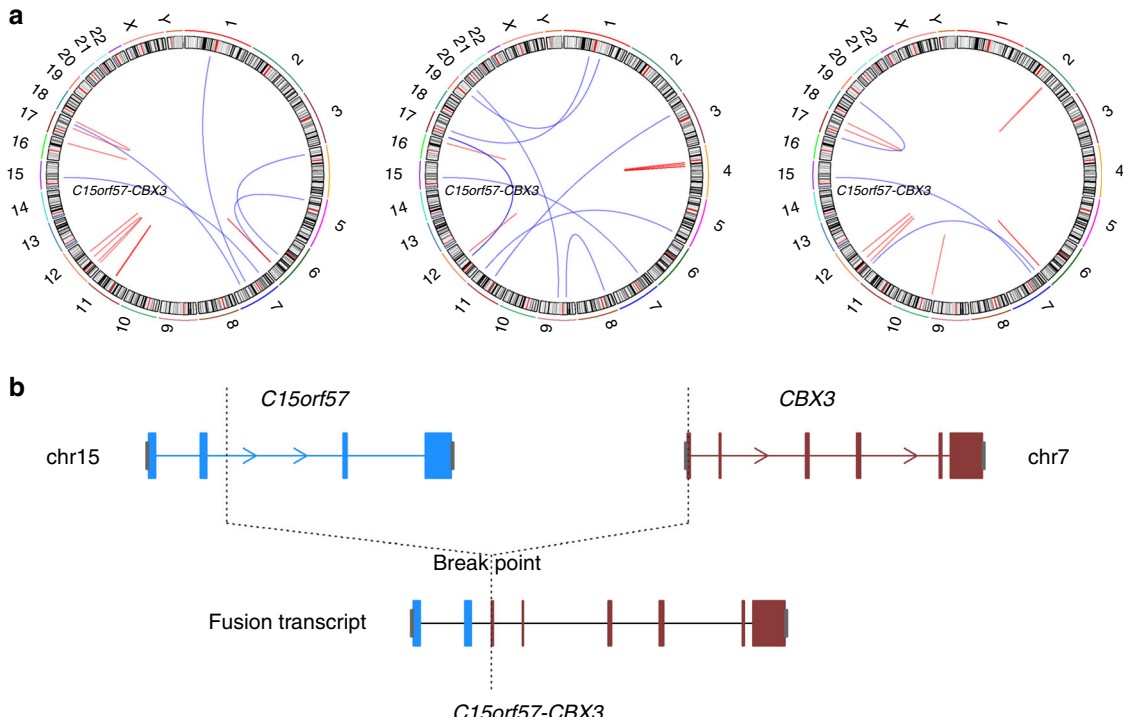

**Fig. 5 Circos plot for sample B972, D739, D740, and schematic sample of fusion genes: *C15orf57-CBX3*. a** Circos plot for Sample B972, D739, D740. *C15orf57-CBX3* was identified from RNAseq. **b** Schematic view of C15orf57-CBX3 fused transcript.

of these genes in pathways and gene ontology (GO) using DAVID[31] (Supplementary Tables 4 and Supplementary Table 5). The top altered pathways and GOs identified in our analyses are related to cell cycle, cell division, or mitosis, showing the potential different mechanisms involved in the tumors in methylation clusters H3-Pons or H3-Medulla. We also applied Gene Set Enrichment Analysis for the normalized FPKM values[32,33] (Fig. 4c–h). Gene sets enriched in methylation cluster H3-Medulla were primarily immune response-related gene sets such as interferon gamma signaling and cytokine receptor interaction, while gene sets enriched in methylation cluster H3-Pons were cell cycle or mitosis-related such as DNA replication, mitotic phase, and checkpoints.

**Fusion genes and copy number alterations**. Using whole genome sequencing data and RNA sequencing data, we evaluated our cohort for genomic rearrangements (Manta[34]) and fusion genes (STAR-fusion[35]). Several common fusion genes were detected, including *KIAA1549/BRAF* ($n = 3$) and *NTRK* fusions, which have been reported in gliomas[11,36,37]. The *KIAA1549/BRAF* fusion was detected in 1 out of 8 pilocytic astrocytomas in our cohort and in two grade II astrocytomas (all 3 in methylation cluster PA-like). We validated several recurrent fusion genes identified in our analyses, including *C15orf57-CBX3* genes ($n = 3$), and *NTRK2*-other genes ($n = 6$) (Fig. 5; Supplementary Table 6 and Supplementary Table 7). Sanger sequencing was performed to confirm the fusion genes and specific breakpoints in these samples.

We also used methylation array data to assess copy number alterations for each methylation cluster by conumee[38] and GISTIC[39] (Supplementary Tables 8 and 9; Supplementary Fig. 7a, b), which showed different patterns in copy number gains and deletions among the four methylation clusters. Although methylation clusters H3-Pons and H3-Medulla shared similar

frequent copy number alterations in 3p26.32, 8p23.1 (gains) and 5q31.3 (loss), H3-Medulla globally harbored more copy number gains (11 loci in H3-Medulla vs. 5 loci in H3-Pons) while H3-Pons exhibited more frequent copy number losses (13 loci in H3-Pons vs. 5 loci in H3-Medulla). Interestingly, only H3-Pons showed 4q12 amplification which contains the frequently amplified gene *PDGFRA* in midline gliomas. Copy number alterations in methylation cluster IDH (7 loci in gains or losses) and PA-like (7 loci in gains or losses, including 7q34: KIAA1549 and BRAF amplification) were fewer in comparison with methylation clusters H3.

**H3-medulla is correlated with better survival than H3-Pons**. We performed survival analysis to investigate potential differences in survival between the distinct methylation clusters we identified (Fig. 6a–d). Kaplan–Meier analyses showed distinct survival curves for patients stratified according to these four methylation clusters (Fig. 6a). Methylation cluster IDH exhibited longer overall survival relative to methylation cluster H3 clusters (Median survival months, IDH: 141.2; H3-Pons: 9.47; H3-Medulla: 26.33; Log-rank test: H3-Pons vs. IDH: $p < 0.0001$; H3-Medulla vs. IDH: $p = 0.0269$). Methylation clusters H3-Medulla and H3-Pons, despite sharing similar genetic alterations of *H3* and *TP53* pathway mutations, had distinct overall survival trends (Log-rank test, $p < 0.0001$) (Fig. 6b).

Cases included in methylation cluster PA-like showed better overall long-term survival compared with the other groups. Importantly, this improved survival trend for patients in the methylation cluster PA-like occurred in the context of the majority of these cases being diagnosed histologically as astrocytoma or oligoastrocytoma, grades II-III (21 out of 34). Only 7 out of 34 tumors in this cluster were diagnosed as pilocytic astrocytoma (Fig. 6a). Collectively, these results suggest that methylation classification into these subgroups may serve as a

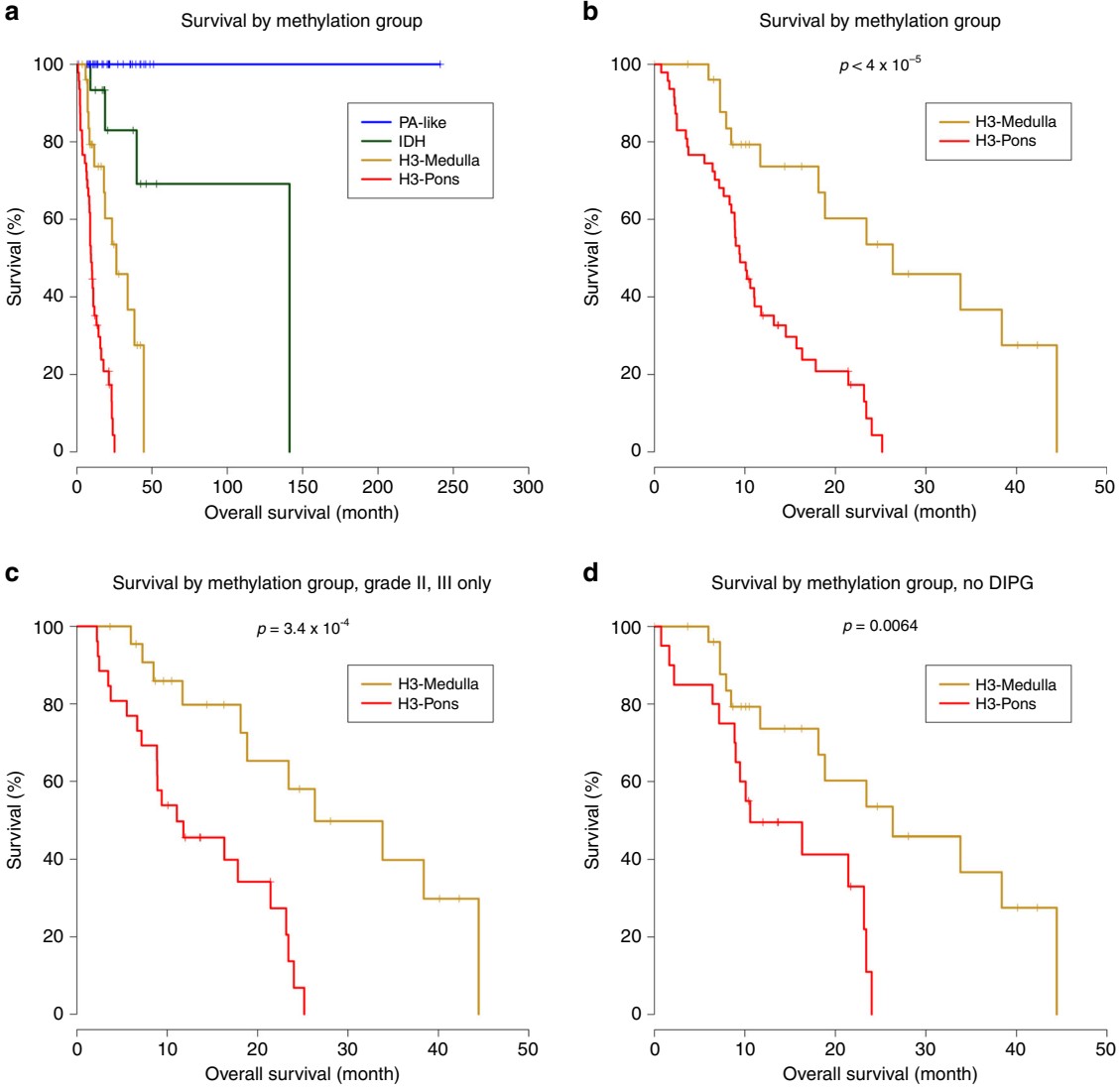

**Fig. 6 Methylation clusters correlate with distinct survival prognosis (Kaplan–Meier curves and log-rank test). a** Survival analysis of brainstem glioma patients for four clusters. **b** Survival analysis of H3-Medulla and H3-Pons only. Log-rank test $p < 4 \times 10^{-5}$ (**c**) Survival analysis of H3-Medulla and H3-Pons, selected samples from tumor of grade II and grade III only. **d** Survival analysis of H3-Medulla and H3-Pons, selected samples from tumors of non-DIPG only.

better correlate to identify patients with grade I–III tumors that have a more benign clinical course.

In terms of tumor grade, most of the tumors from methylation cluster H3-Pons were grade IV (20 out of 47, 42.6%), while 19 tumors were grade III (40.4%), and 8 tumors were grade II (17.0%). Tumors classified in the methylation cluster H3-Medulla were composed of grade II and III cases (II: 11, 40.7%, III: 12, 44.4%;). When evaluating only grade II and III tumors in both groups, H3-Medulla cases exhibited longer median overall survival (26.3 months) compared to tumors classified as H3-Pons (11.1 months) (Log-rank test, $p = 0.00034$) (Fig. 6c). DIPGs are known to have the worst prognosis among brainstem gliomas, and none of the DIPGs in our cohort were in methylation cluster H3-Medulla. When DIPGs in methylation cluster H3-Pons were excluded, samples from methylation cluster H3-Medulla still showed a significantly longer overall survival relative to tumors classified as H3-Pons (26.3 months vs. 10.6 months) ($p$ value = 0.0064, log-rank test) (Fig. 6d). We also conducted Cox proportional hazards regression models for multivariate analysis

(Supplementary Fig. 8). When including methylation cluster and age as factors, H3-Pons still showed higher risk than H3-Medulla (hazard ratio: 1.04–6.6; $p$ value = 0.041), while age showed only limited effect (hazard ratio: 0.95–1.0, $p$ value = 0.066) (Supplementary Fig. 8a). When including whether the sample is DIPG or non-DIPG, methylation cluster remains the most dominant factor (Supplementary Fig. 8b).

## Discussion
Brainstem gliomas represent a heterogeneous group of tumors arising from the midbrain, pons, and the medulla, affecting both children and adults. These tumors have different histologic features, but also differing levels of resectability and therefore variable clinical courses. Among these tumors, DIPG has been the most extensively studied, due to its relative prevalence and lack of therapeutic options resulting in a poor prognosis. Integrated genomic, epigenomic and transcriptomic studies have provided insightful understanding of the tumorigenesis of pediatric DIPG, which also might hold promise for future utilization of molecular

marker-driven clinical trials and use of novel targeted therapies such as HDAC, JMJD3, ACVR1, PPM1D, and BET bromodomain inhibitors, and CDK7 blockade[40–44]. However, the molecular profiling of the many other brainstem tumors that are non-pediatric DIPG has remained elusive due in large part to the lack of sample availability. This has limited our understanding of these diseases and ability to objectively stratify these patients and implement use of novel targeted therapies. Here, we provide a comprehensive integrated genomic analysis of gliomas of the brainstem, with tumors spanning from the most rostral midbrain to the pons and medulla. The newly updated WHO guidelines use a new diagnostic term for brainstem gliomas with the H3 K27M mutation, however, this classification is unable to distinguish the variable prognoses of these gliomas, categorizing all of them as WHO Grade IV. From our study, we show using the epigenetic and genetic signatures that tumors from various locations in the brainstem can be classified into four major epigenetic subtypes, each with distinct clinical courses and potential therapeutic targets.

Using methylation data we showed that brainstem gliomas could be classified into four major methylation clusters: H3-Pons, H3-Medulla, IDH, and PA-like. We summarized the integrated genetic and clinical features of these subtypes in Fig. 7. Our study revealed the presence of two distinct epigenetic subgroups of H3-mutant tumors, H3-Pons and H3-Medulla. There were significant differences in the survival trends between these two clusters, with the H3-Pons group having a more aggressive course as compared to the H3-Medulla tumors. Based on RNA-seq based differential expression analysis, we found these tumors to have different enrichment of gene expression pathways, with the H3-Medulla tumors enriched for immune-response related pathways, as compared with the more aggressive H3-Pons tumors having more cell cycle-related pathways. Despite these tumors having similar mutation patterns, with common alterations in core mutations such as *H3F3A*, these striking differences in epigenetic and expression patterns may inform distinct origins or influences of the tumor microenvironment, and warrant further investigation. These discoveries indicate that methylation status might improve the classification for brainstem gliomas and guide clinical decision making for patients.

Tumors in the PA-like cluster consisted primarily of tumors originally diagnosed as grades II-III infiltrative gliomas, pilocytic astrocytomas, PMA, and PXA. The PA-like-group tumors had a benign clinical course, despite variability in grade. Based on histologic criteria, such variability may lead to classification of a subset of these cases as higher grade gliomas and lead to over-treatment in clinical practice. Here again we demonstrate that epigenetic and genomic patterns can more precisely stratify patient tumors diagnosed with small biopsies. Using methylation-based classification of tumors could better inform clinical decision making and identify patients that are candidates for therapeutic intervention.

We also utilized whole genome sequencing data to establish the mutation landscape of brainstem gliomas and discovered methylation patterns closely matched with mutation landscapes. Several frequently mutated genes were identified in these clusters, including *H3F3A, HIST1H3B, IDH1, TP53, PPM1D, ATM, ATRX, FGFR1, PIK3CA, NF1, PTEN, PDGFRA,* and *TCF12*. We used additional algorithms to predict driver mutations in noncoding regions of genes, such as *UBE3C, CXorf28,* and *EXD3*, as well as structural variants and copy number changes. Although several genes could be identified in multiple clusters, certain genes, such as *NF1* and *PPM1D* were more frequent in cluster H3-Medulla, while the percentage of *TP53* mutations was higher in H3-Pons. Also, we identified the cluster IDH in brainstem gliomas, with tumors in this cluster harboring co-occurring *IDH1, TP53,* and *ATRX* mutations. The majority of these IDH cluster tumors were restricted to adult patients, consistent with previous studies focusing on pediatric brainstem tumors and showing very rare *IDH* mutations in these pediatric tumors. This indicates age is a key factor in developing brainstem glioma with *IDH1* mutation.

This comprehensive study of brainstem gliomas provides an overview of this heterogeneous tumor entity. Using an integrated genomic analysis of more than one hundred brainstem gliomas from various anatomic locations, we show the promise of molecular profiling of brainstem tumors for improved tumor classification and understanding of their molecular underpinnings, and identify new potential therapeutic targets, all to improve outcomes for these patients.

## Methods

**Sample collection and cohort characteristics**. Brain tumor and peripheral blood samples were collected from patients at Beijing Tiantan Hospital, Capital Medical University, China between 2013 and 2017 with informed consent reviewed by Institutional Review Board of Beijing Tiantan Hospital, with accreditation of the Association for the Accreditation of Human Research Protection Program. Biopsy or resected tumors were for clinical diagnosis and therapy. All the FFPE and snap-frozen tumor tissues used for sequencing were reviewed by an experienced team of neuropathologists at Beijing Tiantan Hospital, Captial Medical University. Tissues whose tumor content were less than 70% were excluded from subsequent sequencing. 126 leftover samples were used in this analysis. Among these samples, 97 samples were used for whole genome sequencing, 123 for methylation microarray, 75 for RNAseq, and 21 samples for panel sequencing. Clinical information and survival data are available for these patients. Kaplan–Meier analysis, log-rank test, and Cox proportional hazards regression model (R package survminer) were used to test for survival analysis.

**Whole genome sequencing and RNA sequencing**. Whole exome sequencing, RNAseq, and panel targeted sequencing (for 68 common mutated brain tumor genes) were performed by GenetronHealth, Beijing, China. For whole genome sequencing and panel sequencing, BWA was used for alignment and GATK mutect2 was utilized for variant calling. For RNAseq, STAR, or hisat2 was use for alignment, cufflinks was used for gene expression profiling, Htseq and edgeR were used for differential counts analysis, and GSEA was used for gene sets analysis.

**Methylation microarrays**. The Illumina HumanMethylation450 BeadChip and Infinium MethylationEPIC BeadChip were used for assessing genome-wide methylation profiling of 123 samples. GenomeStudio Methylation Module was used for data processing and quality check. Hierarchical clustering, t-distributed stochastic neighbor embedding (tSNE), and principal component analysis by R package Rtsne, and pheatmap with linkage method WPGMA and Euclidean distance were performed for evaluation of subgroups[18,45–47].

**Copy number alterations**. Segmentation was calculated by Conumee from methylation arrays. GISTIC 2.0 was used for four different methylation clusters. Parameters setting: -genegistic 1 -smallmem 1 -broad 1 -conf 0.95 -armpeel 1 -savegene 1 -gcm mean -maxseg 2500 -ta 0.1 -td 0.1.

**Structural variants**. Manta was applied for structural variant calling from whole genome sequencing data. We also used RNAseq data for checking structural variants, and STAR-fusion was applied for RNAseq data. Selected fusion genes detected from both whole genome sequencing and RNAseq were validated by Sanger sequencing, including: CAPZA2-MET, KMT2E-MET, MET-CTTNBP2, ST7-CAPZA2, CAPZA2-CLCN1, CAPZA2-THAP5, CAPZA2-PNPLA8, RB11-2B6.3-MET, SYS1-DBNDD2, C15orf57-CBX3. Primers for PCR and Sanger sequencing were listed in Supplementary Table 10.

**Reporting summary**. Further information on research design is available in the Nature Research Reporting Summary linked to this article.

## Data availability

Whole Genome Sequencing data of this study has been deposited to Genome Sequence Archive in BIG Data Center, Beijing Institute of Genomics (BIG), Chinese Academy of Sciences[48,49], https://bigd.big.ac.cn/gsa-human, accession number HRA000092, and RNAseq, panel targeted sequencing, and methylation microarray data to the European Genome-phenome Archive (EGA) under accession number EGAS00001004341. The deposited and publicly available data are compliant with the regulations of the Ministry of Science and Technology of the People's Republic of China.

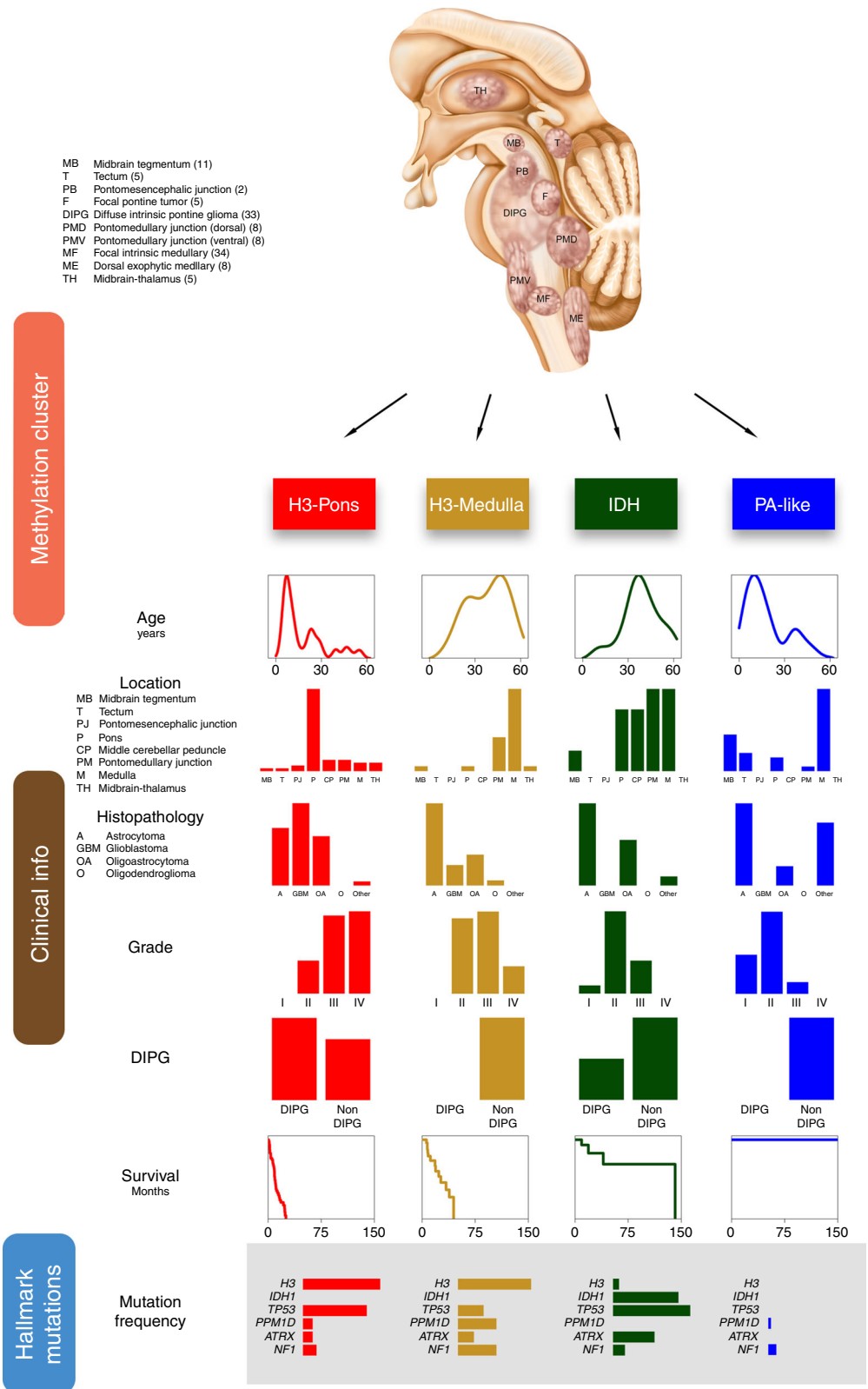

**Fig. 7 Overview of subtypes of brainstem glioma Integrative analysis of brainstem gliomas revealed four different subtypes with distinct clinical, genetic, and epigenetic characteristics.** The illustration on the top shows the locations of tumors selected for this study. Numbers in the parentheses indicated case counts in the code index respectively. Based on methylation patterns, brainstem gliomas could be differentiated into four subtypes: H3-Pons, H3-Medulla, IDH, and PA-like. Clinical information was listed in columns. Tumor location, histopathology, grade, and diagnosis of DIPG are showed in frequency as barplots. Survival data was showed as Kaplan–Meier curves, and age at diagnosis is represented as density plot. Several hallmark mutations are selected and showed in frequency as barplots.

The data will be available for sharing and data use agreements are available in Supplementary materials.

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

## Acknowledgements

The project was supported by National Key Technology Research and Development Program of the Ministry of Science and Technology of China (Grant nos. 2014BAI04B01 and 2015BAI12B04), Beijing Municipal Administration of Hospitals Clinical Medicine Development of Special Funding Support (Grant no. ZYLX201608), the National Natural Science Foundation of China (Grant no. 81872048), Beijing Municipal Natural Science Foundation (Grant no.7161004), 2016 Development Project of Science and Technology Innovation Base (Grant no.206028), and Beijing Municipal Special Funds for Medical Research No. JingYiYan 2018-7 (PI: Liwei Zhang), and by a Duke Brain Tumor Center Grant and Zachem Family Fund (PI: Hai Yan). The authors would like to thank Guilin Li, Lin Luo, Jiang Du, and Junmei Wang (Department of Pathology, Beijing Tiantan Hospital, Capital Medical University) for their assistance in evaluating the histopathological material, Lin Qiao (Beijing Tiantan Hospital, Capital Medical University) for her help in collecting samples, and the research computing facility Duke Compute Cluster (Mark DeLong and Tom Milledge) at Duke University.

## Author contributions

L.H.C., B.H.D., L.Z., and H.Y. conceived and designed this study; L.H.C. and B.H.D developed the methodology; G.L., J.Y., X.W., Ce.X. and S.W. analyzed and interpreted the

data; C.P., Y.Wu., X.C., Y.G., T.S., Y.S., P.Z., Z.W., J.Z., D.L., Y.Z., W.W., Y.Wa., and L.Z. contributed to the acquisition of data; L.H.C. drafted the manuscript; B.H.D., L.J.H., M.S.W., C.P., Ch.X., R.E.M., D.M.A., Y.H., and H.Y. reviewed, and/or revised the manuscript; H.Y. and L.Z. supervised this study.

## Competing interests

H.Y. is the chief scientific officer and has owner interest in Genetron Holdings, and receives royalties from Genetron, Agios, and Personal Genome Diagnostics (PGDX).
