## [Peer Review File · Nature Communications]

Reviewers' comments:

Reviewer #1 (Remarks to the Author):

Using unsupervised analysis of DNA methylation profiles, four midline glioma clusters were identified which tracked with but further sophisticated the now commonly used Heidelberg brain tumor DNA methylation classifier. The clusters showed unique characteristics not only in DNA methylation profile but also in terms of age, molecular alterations, clinical outcome, and site in the midbrain. This is a well written manuscript on a great new dataset and a unique cohort of patients. The further separation in DNA methylation subtypes requires further validation and multivariate testing of outcome differences but may derive clinical relevance. The following comments are intended the help further solidify the manuscript.

1. Patients included in this study were all treated in Beijing; whereas previous reports on midbrain glioma i.e Fontebasso et al, Mackay et al, are likely dominated by Caucasian heritage patients. Can the impact of germline variation be considered in the analysis; are there particular differences in molecular alterations in the current set in comparison to these previous studies, that may tell us something about the interplay between germline and somatic genomic changes?
2. The legend of Figure 1 is nearly unreadable.
3. Mutation count per sample is reported; since the number of mutations detected is also a function of sequence coverage, these numbers should be replaced by the mutation rate per megabase.
4. Are the two H3 clusters (H3-Pons and H3-Medulla) comparable to the two DNA methylation groups identified in PMID 30396367?
5. Given the wide range of ages at diagnosis, the survival curves by itself are difficult to interpret in absence of a multivariate analysis.

Reviewer #2 (Remarks to the Author):

This exciting study provides a well-presented and thorough molecular analysis of a large cohort of rare tumors resulting in important new insights into the heterogeneity and pathogenesis of brainstem gliomas across age groups. The authors use DNA methylation signatures to cluster the tumors into four main groups that are also distinguished by mutation frequencies, age, location and survival. This highlights an opportunity for refined diagnosis, so it is very helpful that the authors compared their methylation classification with that obtained using the online DKFZ classifier which is being adapted by many for diagnostic purposes.

The authors used DNA methylation signature clustering to identify two different subgroups of brainstem glioma with high frequency H3K27M mutation. The differences between these two clusters does not look very robust in the heatmap in Figure 1, and the clusters are not well separated in the PCA analyses in Figure 3. However, one group of tumors was predominantly located in the pons and the other in the medulla, and importantly, the two groups differed in frequency of other mutations, and in age and survival. Therefore, the sample size in this study appears to have uncovered a more refined clustering revealed by sufficient numbers of tumors.

The manuscript would be further strengthened by addressing the following points.

1. Most importantly, the data in this paper is an extremely valuable resource for the worldwide community both for refined tumor classification and for basic research studies of brainstem gliomas. The authors have deposited only the DNA methylation data in GEO. It is critical that the WGS and RNAseq data are also deposited for public access.
2. Please comment whether the samples reviewed by central pathological review. This helps to

interpret the heterogeneity in tumor grade, and the assignments of DIPG vs non-DIPG subgroups in the study.

3. Some additional comment on copy number data would be useful. Were the fusion genes included in regions of genomic amplification? Can genes also be annotated on the GISTIC plots? For example, the focal peak at 4q12 in Supplementary Fig. 6 only appears in the H3-Pons methylation cluster. Presumably this encompasses PDGFRA? Given the frequent SNV or in/del mutation of PDGFRA in this subgroup shown in Fig. 2, this seems a noteworthy distinction for this methylation cluster.

4. Supplementary Table 5 says that genes were selected from Supplementary Table 3 (which contained 20,000 most variable probes in the methylation arrays), but likely means Supplementary Table 4. It is unclear in the table which Methylation clusters are enriched for the terms shown in the table.

5. Supplementary Table 8: A different code was used for the sample names here that cannot be connected to the rest of the supplemental tables or other data in the paper. The sample names should be consistent throughout.

Point-by-Point Response to Reviewers

NCOMMS-19-12395-T

Reviewer notes:

Reviewer #1 (Remarks to the Author):

Using unsupervised analysis of DNA methylation profiles, four midline glioma clusters were identified which tracked with but further sophisticated the now commonly used Heidelberg brain tumor DNA methylation classifier. The clusters showed unique characteristics not only in DNA methylation profile but also in terms of age, molecular alterations, clinical outcome, and site in the midbrain. This is a well written manuscript on a great new dataset and a unique cohort of patients. The further separation in DNA methylation subtypes requires further validation and multivariate testing of outcome differences but may derive clinical relevance. The following comments are intended to help further solidify the manuscript.

1. Patients included in this study were all treated in Beijing; whereas previous reports on midbrain glioma i.e Fontebasso et al, Mackay et al, are likely dominated by Caucasian heritage patients. Can the impact of germline variation be considered in the analysis; are there particular differences in molecular alterations in the current set in comparison to these previous studies, that may tell us something about the interplay between germline and somatic genomic changes?

We appreciate the reviewer's suggestion. To better understand the extent to which our results are informative for classification across ethnicities, we compared our dataset with other studies, eg. Fontebasso et al (Nature Genetics 2014), Sturm et al. (Cancer Cell 2012), TCGA datasets, and Buczkowicz et al. (Nature Genetics 2014). Our initial analyses indicated that the dataset from Fontebasso et al exhibits a more obvious batch effect than the others. Here, we used the combined dataset of our Beijing cohort and data from Buczkowicz et al. to examine the relevance of our identified methylation clusters across datasets with different ethnic representations. All samples in the Buczkowicz study are classified as DIPGs. We used tSNE with top 20000 variable probes from our study on the combined dataset and found that the

DIPG cases from Buczkowicz et al. are closely associated with H3-Pons samples in our cohort, consistent with our observations. Interestingly, all cases from the Buczkowicz et al. study that clustered toward the PA-like group were H3^{WT} and/or previously classified in “silent” or “MYCN” methylation clusters, rather than the H3-K27M group of Buczkowicz et al. Based on these new analyses, we added the following results as highlighted “All patients in this study were of Asian ethnicity. To evaluate if these distinct H3 clusters can be found in a predominantly non-Asian population, we combined our dataset with published studies of 28 DIPG samples (Buczkowicz et al., 2014). From tSNE results of those selected top 20000 variable probes, we found that those DIPG samples grouped closely with our H3-Pons samples as expected (Supplementary Fig. 5), indicating that classification according to the MethylationCluster H3-Pons may be robust across ethnicities. Notably, of the 6 DIPG cases from Buczkowicz et al. that clustered toward the PA-like group, 4 were H3^{WT} and all were previously classified in either the “silent” or “MYCN” methylation clusters of that study.” However, it is important to note that the study did not include medullary tumors, so we cannot comment on the robustness across datasets for this glioma subtype. Overall, the comparison of the data from our cohort with the Buczkowicz et al. study does not reveal any obvious differences in frequency of somatic genetic alterations or presence of methylation signatures.

2. The legend of Figure 1 is nearly unreadable.

We have updated Figure 1.

3. Mutation count per sample is reported; since the number of mutations detected is also a function of sequence coverage, these numbers should be replaced by the mutation rate per megabase.

We thank the reviewers for this suggestion. We have revised Figure 2 to include the values of the mutation rate per megabase.

4. Are the two H3 clusters (H3-Pons and H3-Medulla) comparable to the two DNA methylation groups identified in PMID 30396367?

We thank the reviewing for raising this interesting question. Unfortunately, the referenced study did not deposit the methylation data in a public repository. We have contacted the

corresponding authors several times and have not received any response. Of note, in their study, they focused more on the difference between H3.1 and H3.3 tumors. Most of our H3 mutant samples are H3.3 mutant (only 2 out of our 59 H3 mutant samples are H3.1 mutant). Therefore, we reasoned that by including their samples we would group their H3.3 DIPG samples with our H3-Pons, similar to the observation mentioned in Point #1.

5. Given the wide range of ages at diagnosis, the survival curves by itself are difficult to interpret in absence of a multivariate analysis.

We thank the reviewers for this suggestion. We have added the multivariate analysis and stated in the manuscript as highlighted “We also conducted Cox proportional hazards regression models for multivariate analysis (Supplementary Fig. 8). When including methylation cluster and age as factors, H3-Pons still showed higher risk than H3-Medulla (hazard ratio: 1.04 – 6.6; p-value = 0.041), while age showed only limited effect (hazard ratio: 0.95 – 1.0, p-value = 0.066) (Supplementary Fig. 8a). When including whether the sample is DIPG or non-DIPG, Methylation Cluster remains the most dominant factor (Supplementary Fig. 8b).” and added the figure Supplementary Fig. 8.

Reviewer #2 (Remarks to the Author):

This exciting study provides a well-presented and thorough molecular analysis of a large cohort of rare tumors resulting in important new insights into the heterogeneity and pathogenesis of brainstem gliomas across age groups. The authors use DNA methylation signatures to cluster the tumors into four main groups that are also distinguished by mutation frequencies, age, location and survival. This highlights an opportunity for refined diagnosis, so it is very helpful that the authors compared their methylation classification with that obtained using the online DKFZ classifier which is being adapted by many for diagnostic purposes.

The authors used DNA methylation signature clustering to identify two different subgroups of brainstem glioma with high frequency H3K27M mutation. The differences between these two clusters does not look very robust in the heatmap in Figure 1, and the clusters are not well separated in the PCA analyses in Figure 3. However, one group of

tumors was predominantly located in the pons and the other in the medulla, and importantly, the two groups differed in frequency of other mutations, and in age and survival. Therefore, the sample size in this study appears to have uncovered a more refined clustering revealed by sufficient numbers of tumors.

The manuscript would be further strengthened by addressing the following points.

1. Most importantly, the data in this paper is an extremely valuable resource for the worldwide community both for refined tumor classification and for basic research studies of brainstem gliomas. The authors have deposited only the DNA methylation data in GEO. It is critical that the WGS and RNAseq data are also deposited for public access.

We agree with the reviewer and appreciate this comment. The NGS data, including RNA-seq and WGS, data is now being uploaded to the SRA.

2. Please comment whether the samples reviewed by central pathological review. This helps to interpret the heterogeneity in tumor grade, and the assignments of DIPG vs non-DIPG subgroups in the study.

All the FFPE and snap-frozen tumor tissues used for sequencing were reviewed by an experienced team of neuropathologists Guilin Li, Lin Luo, Jiang Du and Junmei Wang at Beijing Tiantan Hospital. We have now updated our methods to describe the specimen review process and their expertise in neuropathology.

3. Some additional comment on copy number data would be useful. Were the fusion genes included in regions of genomic amplification? Can genes also be annotated on the GISTIC plots? For example, the focal peak at 4q12 in Supplementary Fig. 6 only appears in the H3-Pons methylation cluster. Presumably this encompasses PDGFRA? Given the frequent SNV or in/del mutation of PDGFRA in this subgroup shown in Fig. 2, this seems a noteworthy distinction for this methylation cluster.

We thank the reviewer for raising this interesting point. We would like to include the genes on the GISTIC plots, but with many genes, it becomes difficult to visualize in this format. We have now included the information of genes as new Supplementary Table 9. As for the reviewer's comment of focal peak at 4q12, it indeed encompasses PDGFRA. Also, KIAA1549 and BRAF,

known for fusion gene in pilocytic astrocytoma, were amplified in PA-like methylation cluster. We now added this information in the Result as highlighted “Interestingly, only H3-Pons showed 4q12 amplification which contains the frequently amplified gene *PDGFRA* in midline gliomas.”. We also added the information about KIAA1549 and BRAF amplification as highlighted “including 7q34: KIAA1549 and BRAF amplification”.

4. Supplementary Table 5 says that genes were selected from Supplementary Table 3 (which contained 20,000 most variable probes in the methylation arrays), but likely means Supplementary Table 4. It is unclear in the table which Methylation clusters are enriched for the terms shown in the table.

We apologize for the confusion. We have updated the Supplementary Table 5, showing that genes were selected from Supplementary Table 4. And we also updated Supplementary Table 5 to clarify which Methylation clusters are enriched.

5. Supplementary Table 8: A different code was used for the sample names here that cannot be connected to the rest of the supplemental tables or other data in the paper. The sample names should be consistent throughout.

We apologize for this inconsistency. We have updated the codes for the sample names in Supplementary Table 8.

REVIEWERS' COMMENTS:

Reviewer #1 (Remarks to the Author):

The authors have addressed the minor concerns raised on the initial submission and I support publication.

Reviewer #3 (Remarks to the Author):

The authors addressed all of the questions raised in the prior review. The manuscript makes a significant contribution to the molecular characterization of brainstem gliomas across age groups.